# Allelic Expression Imbalance Analysis Identified *YAP1* Amplification in p53- Dependent Osteosarcoma

**DOI:** 10.3390/cancers13061364

**Published:** 2021-03-18

**Authors:** Guanglin Niu, Agnieszka Bak, Melanie Nusselt, Yue Zhang, Hubert Pausch, Tatiana Flisikowska, Angelika E. Schnieke, Krzysztof Flisikowski

**Affiliations:** 1Chair of Livestock Biotechnology, Technical University of Munich, 85354 Freising, Germany; niu@wzw.tum.de (G.N.); agnieszka.bak@wzw.tum.de (A.B.); melanie.manyet@tum.de (M.N.); yuezhang@wzw.tum.de (Y.Z.); tatiana.adamowicz@wzw.tum.de (T.F.); schnieke@wzw.tum.de (A.E.S.); 2Institute of Agricultural Sciences, ETH Zurich, 8092 Zurich, Switzerland; hubert.pausch@usys.ethz.ch

**Keywords:** *YAP1*, *TP53*, *TP63*, *p16*, allelic expression imbalance, copy number variation, bone cancer, pig

## Abstract

**Simple Summary:**

Osteosarcoma (OS) is a highly heterogenous cancer, making the identification of genetic driving factors difficult. Genetic factors, such as heritable mutations of *Rb1* and *TP53,* are associated with an increased risk of OS. We previously generated pigs carrying a mutated *TP53* gene, which develop OS at high frequency. RNA sequencing and allelic expression imbalance analysis identified an amplification of *YAP1* involved in p53- dependent OS progression. The inactivation of *YAP1* inhibits proliferation, migration, and invasion, and leads to the silencing of *TP63* and reconstruction of *p16* expression in p53-deficient porcine OS cells. This study confirms the importance of p53/YAP1 network in cancer.

**Abstract:**

Osteosarcoma (OS) is a primary bone malignancy that mainly occurs during adolescent growth, suggesting that bone growth plays an important role in the aetiology of the disease. Genetic factors, such as heritable mutations of *Rb1* and *TP53,* are associated with an increased risk of OS. Identifying driver mutations for OS has been challenging due to the complexity of bone growth-related pathways and the extensive intra-tumoral heterogeneity of this cancer. We previously generated pigs carrying a mutated *TP53* gene, which develop OS at high frequency. RNA sequencing and allele expression imbalance (AEI) analysis of OS and matched healthy control samples revealed a highly significant AEI (*p* = 2.14 × 10^−39^) for SNPs in the *BIRC3-YAP1* locus on pig chromosome 9. Analysis of copy number variation showed that *YAP1* amplification is associated with the AEI and the progression of OS. Accordingly, the inactivation of *YAP1* inhibits proliferation, migration, and invasion, and leads to the silencing of *TP63* and reconstruction of *p16* expression in p53-deficient porcine OS cells. Increased *p16* mRNA expression correlated with lower methylation of its promoter. Altogether, our study provides molecular evidence for the role of *YAP1* amplification in the progression of p53-dependent OS.

## 1. Introduction

Osteosarcoma (OS) is the most common form of primary bone cancer, mainly found in young people with the second highest incidence group being over the age of 50 [1]. OS is a highly aggressive cancer for which there have been no major therapeutic improvements over the last decades [2,3]. The five-year overall survival rates are 69% and 23% for primary and metastatic OS respectively [4]. Risk factors for OS include rapid bone growth [5], gender [6], a tall stature [6], and radiation as a source for secondary OS [7,8].

High heterogeneity of OS hinders the identification of driver mutations and therapeutic target genes. Genome-wide somatic alterations in OS include mutations and chromosomal lesions, such as structural and copy number variations in *TP53* [9,10,11] and *Rb1* [9,12], alternative lengthening of telomeres, and an aberrant epigenetic pattern [9,13,14,15,16,17,18].

Germline *TP53* mutations predispose for developing different types of cancers including OS in Li- Fraumeni syndrome patients [19,20]. In mice, p53-deficiency leads to OS at low frequency, however conditional activation of p53 mutation such as R172H in osteoblasts has increased the aggressiveness of the disease [21,22].

We have generated genetically engineered pigs with a Cre-inducible oncogenic *flTP53^R167H^* mutation, homologous to the human hotspot mutation *TP53^R175H^*, a structural mutation with oncogenic function [23,24]. The *flTP53^R167H^* heterozygous and homozygous pigs carrying the uninduced allele frequently develop OS in long bones and skull. The porcine OS showed similar features to human OS, such as tumour cells with a highly abnormal karyotype, nuclei with atypical mitotic figures, and increased resistance to radiation [25].

To identify the genetic factors which contribute to the OS development in the *flTP53^R167H^* pigs, we have carried out RNA sequencing and allele expression imbalance (AEI) analysis. The AEI quantifies the stoichiometric difference in the expression of the two alleles of a genetic locus [26,27,28] or two haplotypes of a diploid individual which can be distinguished at heterozygous variation sites [29]. Compared to gene expression analysis, AEI has the advantage of using two alleles of one gene within individuals and thus better controlling the genetic background and environmental effects, and therefore can sensitively and accurately detect the genetic and epigenetic differences in highly heterocellular samples such as tumors [30,31]. AEI has been applied to detect driver mutations in various human cancers including colorectal and breast cancer [32,33].

## 2. Material and Methods

### 2.1. Animals

OS (*n =* 48) samples were collected from ten (8 male and 2 female) fl*TP53^R167H/R167H^* reached sexual mature and 24 (13 male and 11 female) fl*TP53^R167H/+^* pigs aged 7–32 months. All animal experiments were approved by the Government of Upper Bavaria (permit number 55.2-1-54-2532-6-13) and performed according to the German Animal Welfare Act and European Union Normative for Care and Use of Experimental Animals.

### 2.2. Necropsy Examination and Tumour Analysis

Pigs were humanely killed and examined by complete necropsy at the Tiergesundheitsdienst Bayern (Bavarian Animal Health Service). In total, 48 OS and matched healthy bone samples from hetero- and homozygous fl*TP53^R167H^* pigs were analysed, as previously described [34].

### 2.3. Next-Generation RNA Sequencing

Briefly, ten mg of OS and matched healthy samples was used for total RNA extraction using Zymo Direct-zol RNA MiniPrep kit (Zymo Research, Freiburg, Germany) according to the manufacturer’s instructions. The quality and quantity of RNA samples was measured using an Agilent RNA 6000 Nano kit (Agilent, Waldbronn, Germany) on a 2100 Bioanalyzer (Agilent) and a Nanodrop 2000 spectrophotometer (Thermo Scientific, Waltham, MA, USA). The RNA integrity values (RIN) ranged from 7.6 to 9.0. 400 ng total RNA was used for library preparation with the TruSeq RNA Library Preparation Kit v2 (Illumina, San Diego, CA, USA) according to the manufacturer’s instructions, as described in our earlier study [35]. Libraries were sequenced with a HiSeq2500 sequencing system (Illumina) to produce 100-base-paired end reads for 17 samples. An average of 56 million reads per sample was generated. Reads were pseudoaligned against an index of the porcine transcriptome (Sscrofa 11.1; Ensembl release 91, Hinxton, UK) and quantified using *kallisto* (version 0.43.1, Nicolas L Bray et al., Pasadena, CA, USA) [36]. The differential expression of transcripts was quantified using a likelihood-ratio test implemented in the R package sleuth (version 0.29.0, Hadley Wickham and Jenny Bryan, Auckland, New Zealand) [37]. Hierarchical clusters and heat maps for genes with the most pronounced different levels of expression were generated using the heatmap.2-function of the R package gplots [38].

For allele expression imbalance analysis, variant calling based on STAR alignments was performed according to GATK [39] best practice recommendations for RNAseq [40,41]. The GATK tool Split N Cigar Reads was used to split reads into exons and remove false variants resulting from overhangs. This step included reassignment of the STAR alignment mapping qualities. GATK recalibration of base scores was based on the Ensembl release 83 variant database. Variant calling was carried out using GATK Haplotype Caller with the don’t Used Soft Clipped Bases option. GATK Variant Filtration was applied to clusters of at least 3 SNPs within a window of 35 bases between them with the following parameters: Fisher strand value (FS) > 30.0 and a quality by depth value (QD) < 2.0. The probability of allelic imbalance for each SNP was calculated based on the number of references and alternate allele reads in heterozygous animals using a two-sided binomial test. *p* values were adjusted for false discovery rate (q value) to take account of multiple testing.

### 2.4. Gene Set Enrichment Analysis

Gene set enrichment analysis was performed using GSEA software (version 2.2.4, Tamayo et al., San Diego, CA, USA) [42]. The log2 fold change, adjusted *p*-value, and the Human Genome Organisation (HUGO) gene symbols were used to generate a pre-ranked file as input for the GSEA Preranked tool. Enrichment analysis was performed using the following specifications: classic enrichment statistics, 1000 permutations and hallmark gene sets from Molecular Signatures Database (MSigDB) (version 6.1, Tamayo et al., San Diego, CA, USA) [43].

### 2.5. Quantitative Real-Time RT-PCR

Breifly, 200 ng total RNA was used for cDNA synthesis using the Superscript IV (Thermo Fisher, Waltham, MA, USA) according to the manufacturer’s instructions. The detailed description of the qRT-PCR was previously described [35]. The relative expression was normalised to *GAPDH* expression and statistically compared using Students *t*-test. All primer sequences used in the study are shown in Appendix A.

### 2.6. Pyrosequencing

Pyrosequencing assays were designed using PyroMark Assay Design 2.0 software (Qiagen, Düsseldorf, Germany). Thereby, 500 ng genomic DNA was bisulfite-converted with the EZ DNA Methylation-Direct kit (Zymo Research, Irvine, CA, USA) according to the manufacturer’s instructions. A detailed description of pyrosequencing was previously provided [35].

### 2.7. Droplet Digital PCR (ddPCR)

Genomic DNA was digested with HindIII (NEB, Frankfurt am Main, Germany) using 3 U/μg DNA. The detailed description of the ddPCR was previously described [44]. Reagents and equipment were from Bio-Rad Laboratories (Hercules, CA, USA) unless otherwise specified.

*YAP1* promoter copy number was determined using the fluorescence-labelled YAP1-1 probe (5′FAM-cgcgggagggtttaagtgg-BHQ3′) and primers YAP1-F1 (5′-tgttacaggtaccattgtgctcca-3′) and YAP1-R1 (5′-cagtccccgggaaaggttg-3′) amplifying a 182 bp fragment. *YAP1* exon 2 copy number was determined using the fluorescence labelled YAP1-2 probe (5′FAM-ttctagcgtttgcaaacata-BHQ3′) with primers YAP1-F2 (5′-agataacataggataggtct-3′) and YAP1-R2 (5′-tgcagagaatgcatagttt-3′) amplifying a 147 bp fragment. YAP1- 3′UTR copy number was determined using the fluorescence-labelled YAP1-3 probe (5′FAM-ttgcgaccttctggccaata-BHQ3′) and primers YAP1-F3 (5′-ccctcaggtagactgcattc-3′) and YAP1-R3 (5′-gaaagaatcttgctggacgtt-3′) amplifying a 138 bp fragment. Porcine *GAPDH* was used as reference [44]. Primers and probes were from Eurofins Genomic.

### 2.8. Western Analysis

Protein was isolated using T-PER Tissue Protein Extraction Reagent (Thermo Scientific) and Western analysis was carried out using iBind Western System (Thermo Scientific) according to the manufacturers’ instructions. Pig YAP1 was detected using polyclonal rabbit anti-YAP1 ARP50530 (diluted 1:1000) and horseradish peroxidase (HRP) labelled anti-rabbit sc-2004 (diluted 1:2000). GAPDH was detected using mouse monoclonal anti-GAPDH #G8795 (diluted 1:3000) and rabbit anti-mouse IgG H&L (HRP) ab6728 (diluted 1:5000).

### 2.9. Immunohistochemistry

Immunohistochemistry was performed as described previously [44]. OS samples (*n =* 6) from *flTP53^R167H^* homozygous pigs were fixed in 4% formalin and decalcified in Osteosoft^®^ (Merck, Darmstadt, Germany). Four-micrometer sections were air-dried for 10 min at 60 °C on a glass slide. Antigen demasking was performed using the heat retrieval procedure (20 min, citrate buffer pH 6, pressure cooker in microwave medium intensity). Sections were stained with biotinylated rabbit anti-YAP1 antibody (diluted 1:200; ARP50530_P050, Aviva System Biology Cooperation, San Diego, CA, USA) and binding visualized with the avidin-peroxidase solution (ABC kit, Vector, Darmstadt, Germany) followed by DAB staining (Vector). Sections were lightly counterstained with haematoxylin (Merc, Darmstadt, Germany). Pig duodenum sections were used as a positive control. No incubation with primary antibody was used as a negative control.

### 2.10. Generation of sgRNA Construct

SgRNA construct targeting the ATG site in pig *YAP1* was generated by cloning the gRNA oligonucleotides (gRNA_YAP1_1F:5′-GAGGCAGAAACCATGGATCC-3′; gRNA_YAP1_1R: 5′-GGATCCATGGTTTCTGCCTC-3′) into pX330-U6-Chimeric_BB-CBh- SpCas9 vector (Addgene plasmid # 42230; http:/n2t.net/addgene:4223O;RRID:Addgene_42230, accessed on 25 September 2020), which was digested with BbsI restriction enzyme, and cotransfected into pig *flTP53^R167H^* OS cells.

### 2.11. Immunofluorescence Assay

Porcine OS cells (*YAP1*^−^*^/^*^−^*/flTP53^R167H^* and *flTP53^R167H^*/GFP control) were plated on 6 well plates, cultivated till 80% confluency. For fixation, the cells were washed twice with PBS and incubated for 15 min at room temperature in Fixative. Afterward, the cells were washed two times with TBST, permeabilised for 20 min at room temperature with permeabilisation buffer, and blocked for 60 min with 5% BSA. The primary Ki67 antibody (diluted 1:200, MA5-14520, Invitrogen, Waltham, CA, USA) was incubated at 4 °C overnight, afterwards the secondary antibody (Goat Anti-rabbit IgG (H+L) Alexa Fluor Plus 488, diluted 1:300; A32731, Invitrogen) was added and incubated for 60 min at room temperature. 300 nM DAPI (D9564, Sigma) was incubated for 10 min at room temperature (protected from the light). The signal detection was performed under a fluorescence microscope.

### 2.12. Proliferation Assay

Porcine *flTP53^R167H^* OS cells were transfected with gRNA_YAP1 construct by electroporation using the EMC830 electroporation system (BTX). GFP vector was used as a control. Cells were selected by using 200 ng/μL of puromycin for 2 days. After selection and single-cell clone picking, 5 × 10^5^
*YAP1^−/−^/flTP53^R167H^* OS cells were plated on 6-well plates (3 times for each assay). Cells were counted after 24 h, 48 h, 72 h, 96 h, and 120 h of incubation using an automated cell counter (Invitrogen).

### 2.13. Migration and Invasion Assay

For the migration assays, 1 × 10^5^ OS cells (GFP control, *YAP1^−/−^/flTP53^R167H^*) were plated in the upper chambers of 24-well 8.0 μm transwell inserts (Corning Inc.). For invasion assay, 1 × 10^5^ cells were plated in 10% Matrigel-coated 24-well 8.0 μm transwell inserts (Corning Inc, Corning, NK, USA) Medium with FCS was added at the bottom of each transwell. For migration and invasion, chambers were incubated for 24 h. The medium was removed from 24-well plates and transwell inserts, cells were fixed with methanol and stained with Crystal Violet, washed six times with water, and air-dried overnight. Images were taken by a microscope camera, migration/invasion, and the total number of cells was counted. Each experiment was conducted in triplicate.

### 2.14. Availability of Data and Materials

This study utilised porcine reference genome 11.1 assembly which is publicly available from the NCBI assembly database.

The RNA sequencing data has been deposited at the European Nucleotide Archive (ENA) of EMBL-EBI (https://www.ebi.ac.uk/ena/, accessed on 24 February 2019) under primary accession number PRJEB30086. All other data generated during this study are included in this article and its additional files.

## 3. Results

### 3.1. Frequency of OS in flTP53^R167H^ Pigs

A total of 39 fl*TP53^R167H/+^* heterozygous and ten fl*TP53^R167H/R167H^* homozygous pigs were examined by necropsy. Out of 29 heterozygous animals, 18 developed bone tumours by the age of 36 months and 10 out of 10 homozygous pigs by the age of 16 months, all of which were classified histologically as osteoblastic osteosarcomas, as previously described [34].

### 3.2. Genome-Wide Allelic Expression Imbalance Analysis of OS

To identify the transcriptome changes in OS, we first compared RNA sequencing data between OS (*n =* 8) and matched healthy bones from *flTP53^R167H/R167H^* homozygous pigs. This analysis didn’t identify any significantly differentially expressed genes (DEGs) after multi-comparison testing (Appendix A). We concluded that the high heterogeneity of OS limited the identification of DEGs and therefore decided to perform allele expression imbalance (AEI) analysis.

In total, we identified 9657 heterozygous SNPs, of which 144 (*p* < 5.18 × 10^−6^, Bonferroni-adjusted significance threshold) showed AEI in OS samples. The most significant AEI (*p* < 2.73 × 10^−19^) was found for eight SNPs located on pig chromosomes 6, 9, 14 and 16 (Figure 1A, Table 1). Of these, the SNP:33044172A/G located in the 3′UTR of *BIRC3* (Figure 1B) showed the greatest AEI = 0.78 in OS (*p* = 2.14 × 10^−39^). In the neighbourhood of *BIRC3* is *YAP1*; both genes are transcribed in the sense direction. *YAP1* is an evolutionary conserved transcription cofactor of the Hippo pathway, which regulates the development of organs and is deregulated in many human cancers [45]. Because of the functional relevance, our further analyses focused on the *BIRC3-YAP1* locus. By pyrosequencing, we confirmed AEI for the SNP:33044172A/G (0.74 ± 0.12 vs. 0.4 ± 0.05; *p* = 1.35 × 10^−9^; Figure 1C) in a larger cohort (*n =* 48) of OS samples. Next, we aimed to identify the underlying mechanism by which OS achieved the AEI at this SNP. AEI might be linked to DNA polymorphism or associated with DNA methylation in regulatory regions, or resulting from copy number variations (CNV) in the target gene. The presence of DNA regulatory polymorphisms was searched in *BIRC3* (−2000 bp upstream of transcription start site (TSS) on GenBank sequence XM_013979324) and *YAP1* (GenBank XM_021062706) promoter regions in OS (*n =* 8) and adjacent healthy bone samples from *flTP53^R167H/R167H^* homozygous pigs. No polymorphism linked to the AEI of 9:33044172 SNP in the analysed promoter regions was found. Subsequently, CpG islands (CGI) (−379 bp to +42 bp) and (–297 bp and +65 bp) were identified in the promoters of *BIRC3* and *YAP1,* respectively. Pyrosequencing of five CpG sites at *BIRC3* CGI and eight CpG sites at *YAP1* CGI revealed similar DNA methylation (<10% for *BIRC3* and <5% for *YAP1*) in OS (*n =* 48) and matched healthy bone samples from *flTP53^R167H^* pigs (Appendix A). Moreover, the presence of CNVs was analysed using three digital droplet PCR (ddPCR) probes hybridising to the promoter, internal exon and 3′UTR in both genes. No CNV in *BIRC3* in analysed OS (*n =* 48) was found.

In the *YAP1*, copy number ranged from 1 to 68 and was highly associated (*n =* 48; *p* = 1.76 × 10^−8^) with the AEI of 9:33044172A/G SNP (Figure 2A). The increased copy number of *YAP1* correlated with higher expression of the 9:33044172 A allele. Two copies of the *YAP1* were observed in wild-type and healthy bone samples of *flTP53^R167H^* pigs. The expression of the 9:33044172 A allele (Figure 2B) and copy number of *YAP1* (Figure 2C) were positively correlated with the size (ranged 1.5–18 cm) of pig OS. Quantitative PCR (qPCR) and western blot analyses revealed an increased mRNA (*n =* 48; 1.8- fold, *p* < 0.01; Figure 2D) and protein (*n =* 16; 7.7- fold, *p* < 0.0001; Figure 2E) expression in OS compared to adjacent healthy bone samples. The immunostaining showed that YAP1 protein is expressed in the nucleus of pig OS (Figure 2F, Appendix A). Remarkable, no significant differential *YAP1* mRNA expression in the RNA sequencing study was found. A detailed analysis of RNA sequencing data showed only 1.8- fold expression difference of *YAP1* between the OS and healthy bone samples, which was below the two-fold threshold applied.

Together, these results suggested that the *YAP1* amplification is responsible for the AEI of 9:33044172 A/G SNP and plays an important role in the growth of OS in *flTP53^R167H^* pigs.

### 3.3. YAP1 Deficiency Affects the Functional Properties of flTP53^R167H^ OS Cells

To explore the functional impact of *YAP1* knockout on p53-dependent bone tumorigenesis, a CRISPR-Cas9 system was used for targeting the ATG site of *YAP1* in OS primary cells from *flTP53^R167H^* homozygous pigs. Sanger sequencing of the edited OS cells revealed a deletion of 22 nucleotides (+1bp to 22bp; GenBank NC_010451) (Figure 3A), which resulted in a loss of YAP1 expression (Figure 3B). The *YAP1^−/−^/flTP53^R167H^* OS cells showed rounded cell morphology (Figure 3C) compared to normal spindle-shaped OS morphology, reduced proliferation (Figure 3D), migration (Figure 3E,F), invasion (Figure 3E,G), and Ki67 expression (Figure 3H,I) compared to *flTP53^R167H^* OS cells. These data confirmed the importance of YAP1 expression for the p53-dependent progression of OS.

### 3.4. YAP1 Deficiency Leads to Upregulation of p16 and Rb1 and Downregulation of TP63 in flTP53^R167H^ OS Cells

Further, the effect of *YAP1* deficiency on the expression of tumorigenesis-related genes (*Rb1*, *WRAP53, TP53INP1*, *p14*, *p16*, *TP63*, *TP73*) was studied. RT-PCR and qRT-PCR analyses revealed upregulation of *p16* and *Rb1* (2- fold) and downregulation of *TP63* in *YAP1^−/−^/flTP53^R167H^* compared to *flTP53^R167H^* pig OS cells (Figure 4A,B). The downregulation of p63 was confirmed by Western blot (Figure 4C). In summary, the *flTP53^R167H^* OS cells expressed *TP63* but not *p16*, while the knockout of YAP1 resulted in the silencing of p63 and reconstruction of p16 (Figure 4A,C).

We then investigated whether the upregulation of *p16* and *Rb1* in *YAP1^−/−^/flTP53^R167H^* primary OS cells is associated with DNA methylation in the promoter regions of these genes. Using in silico analysis, genomic regions up to −2000bp from the putative transcription start site (TSS) were analysed and CGI at position −132bp to +108bp (GenBank NC_010443) in *p16*, and at position −210bp to −65bp (GenBank NC_010453) in *Rb1* promoters were identified. Within these CGIs, methylation at 8 CpG sites in *p16* and 9 CpG sites in *Rb1* promoters in *YAP1^−/−^/flTP53^R167H^* (*n =* 3) and *flTP53^R167H^* pig OS (*n =* 3) cell lines was measured by pyrosequencing. For *p16*, 6 of 8 CpG sites showed significantly (*p* < 0.05) lower methylation in *YAP1^−/−^/flTP53^R167H^* compared to *flTP53^R167H^* pig OS cells (Figure 5A). Specifically, lower methylation was found at the CpG2 site (25% vs. 33%), CpG3 (6% vs. 59%), CpG4 (17% v 49%), CpG6 (42% vs. 49%), CpG7 (27% vs. 83%), and CpG8 (57% vs. 91%). For *Rb1*, the observed methylation differences were only slightly lower in *YAP1^−/−^/flTP53^R167H^* compared to *flTP53^R167H^* pig OS cells (Figure 5B). The methylation of the *p16* promoter appeared to be correlated with its gene expression in *YAP1^−/−^/flTP53^R167H^* OS cells.

## 4. Discussion

Several next-generation sequencing studies have been performed to identify driver mutations in human OS [9,46,47]. These studies have detected either well-known cancer driver genes such as *TP53*, *RB1*, *BRCA1*, *PTEN*, *ATRX,* or likely passenger mutations [48].

In this work, we utilised the allele-specific expression analysis to reduce the impact of the tumoral heterogeneity on the gene expression and demonstrated the role of *YAP1* amplification in OS progression in p53-deficient pigs. The *YAP1* amplification led to an overexpression of nuclear YAP1 and correlated with OS progression. This finding is consistent with studies showing an increased expression of YAP1 associated with poor prognosis and chemical resistance in human OS [49,50]. Notably, the downregulation of YAP1 reduced the oncogenic potential of human OS cells [16,51].

Multiple mechanisms, such as the TEAD1 signalling pathway [52], an overexpression of Hedgehog (Hh) [51], suppression by miR-1285-3p [53], and circFAT1 [54] are involved in the regulation of YAP1 in tumours. Here, we show that *YAP1* expression is associated with its gene amplification in p53-dependent OS. The amplification of *YAP1* has been detected in different cancers including medulloblastoma [55], metastatic brain cancer [56] and oesophageal squamous cell carcinoma [57]. The coregulation of p53 and YAP1 has been reported in pancreatic cancer (PDAC) where p53-deficiency promoted YAP1 activity [58,59]. Moreover, YAP1 deletion blocked PDAC initiation driven by KRAS and p53 mutations [60]. In this study, we showed that YAP1 deficiency reduces the tumorigenic potential of p53 deficient OS cells.

We recently identified a mutant R167H-Δ152p53 isoform in *flTP53^R167H^* pigs, which is overexpressed in OS [34]. The cooperative role for the p53 pathway and YAP1 in mediating the tumorigenesis has been reported, reviewed in [61]. YAP1 interacts with mutant p53, including the R175H mutation and induces the expression of several pro-oncogenic genes [62]. Also, the nuclear localisation and activity of YAP1 are dependent on p53. While tumours with wild-type *TP53* showed a lack of YAP1 nuclear localisation in pancreatic cancer [58], a loss of p53 in mutant KRAS^G12D^ lung cancer leads to increased YAP1 nuclear localisation [63]. We showed a high activity of nuclear YAP1 in OS from *flTP53^R167H^* pigs. Importantly, the nuclear YAP1 localisation is negatively associated with survival in OS patients [50]. These findings suggest an interaction between mutant p53 isoform and YAP1 in pig OS.

YAP1 physically interacts with p53 family members, p63, p73 [61], and regulates the p53/Rb1/p16 dependent cellular senescence [64]. The ΔNp63 isoform regulates translocation of YAP1 in squamous carcinoma [65] and in response to DNA damage. YAP1 functions also as a transcriptional coactivator of p73-mediated apoptosis [66]. In line with this data, we found that knockout of YAP1 silences p63 and upregulates *p16* in p53-deficient OS cells. In addition, YAP1 deletion reduced the *p16* promoter methylation. YAP1 functions as a key transcriptional regulator of multiple metabolic pathways including the synthesis of compounds such as S-adenosyl methionine (SAM) needed for DNA methylation. A previous study showed that YAP1 deletion downregulated SAM in primary pancreatic tumour cells [59].

## 5. Conclusions

Given the difficulties in restoring the wild-type function of p53 in cancer, YAP1 is potentially the central target for drug development to treat the oncogenic YAP/Hippo-p53 signalling. However, the complex interaction between p53 family members and YAP/Hippo pathway is still not fully understood. Recent studies have proven the importance of genetically engineered pigs as an animal model in oncology [67,68,69,70]. By using *flTP53^R167H^* pigs, we identified the role of YAP1 in the progression of p53- dependent OS. This study confirms the importance of p53/YAP1 network in cancer.

## Figures and Tables

**Figure 1 cancers-13-01364-f001:**
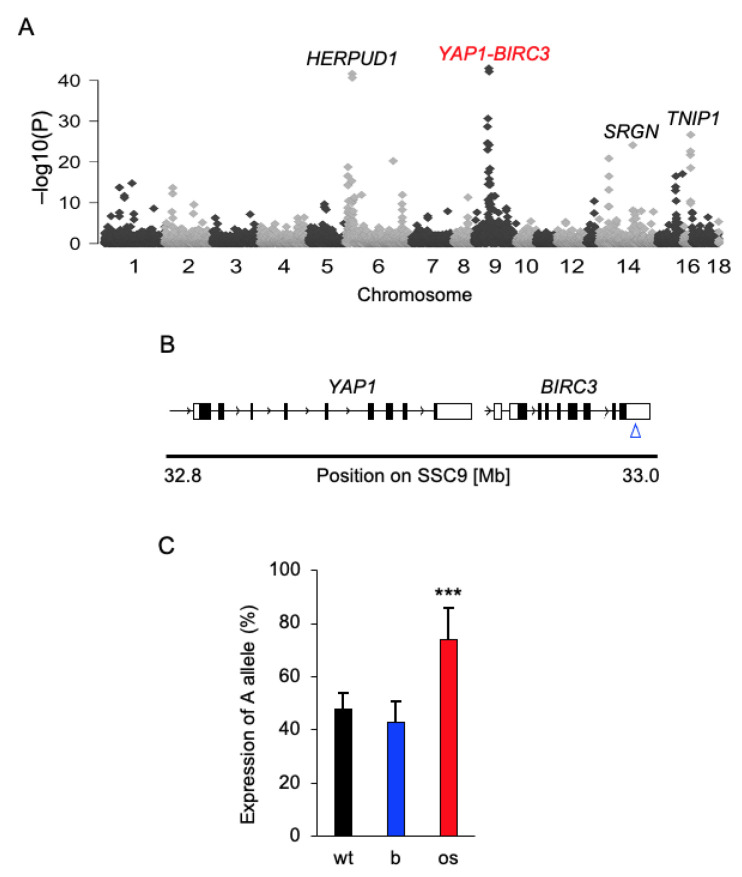
Genome-wide allelic expression imbalance analysis in osteosarcoma. (**A**) A Manhattan plot showing the result of the genome-wide allelic imbalance analysis. The most significant SNPs were found on pig chromosomes 6, 9, 14 and 16. (**B**) Schematic genomic structure of the *YAP1-BIRC3* locus on chromosome 9 in pigs. The blue arrow indicates the position of the 9:33044172 A/G SNP in the 3′UTR of *BIRC3*. (**C**) cDNA pyrosequencing result for the SNP 9:33044172 A/G in osteosarcoma (os, *n =* 48) and matched healthy bone (b) samples collected from *flTP53^R167H^* pigs. To test analysis the validity of the pyrosequencing assay, we used DNA samples (*n =* 5) extracted from wild-type pigs. *** *p* < 0.001.

**Figure 2 cancers-13-01364-f002:**
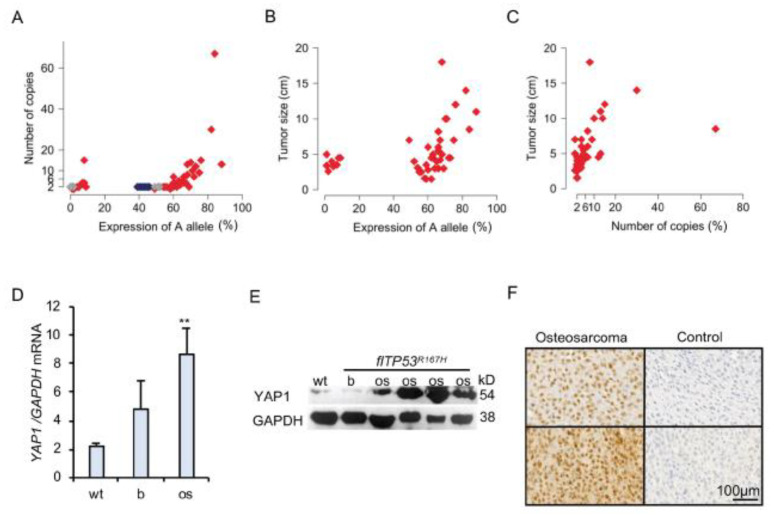
*YAP1* amplification in p53 deficient osteosarcoma. (**A**) Point plot showing the correlation between 9:33044172 A allele expression and *YAP1* copy number. Gray and red points show expression of A allele in bone and OS samples, respectively. Blue points show the measurements in wild-type samples. (**B**) Point plot showing the correlation between 9:33044172 A allele expression and OS (*n =* 48) size. (**C**) Point plot showing the correlation between *YAP1* copy number and OS (*n =* 48) size. (**D**) Quantitative PCR of *YAP1* mRNA expression in wild type (wt, *n =* 5) bones, as well as OS (*n =* 48) and matched healthy bone samples from *flTP53^R167H^* pigs. (**E**) Representative Western blot showing YAP1 expression in wild type bone, OS and healthy bone samples from *flTP53^R167H^* pigs. The uncropped Western blots have been shown in Appendix A. (**F**) Immunohistochemistry staining showing the nuclear location of YAP1 in sections of osteosarcoma from *flTP53^R167H^* pigs. Control samples show staining without the first antibody. Scale bars- 100 μm. (** *p* < 0.01)

**Figure 3 cancers-13-01364-f003:**
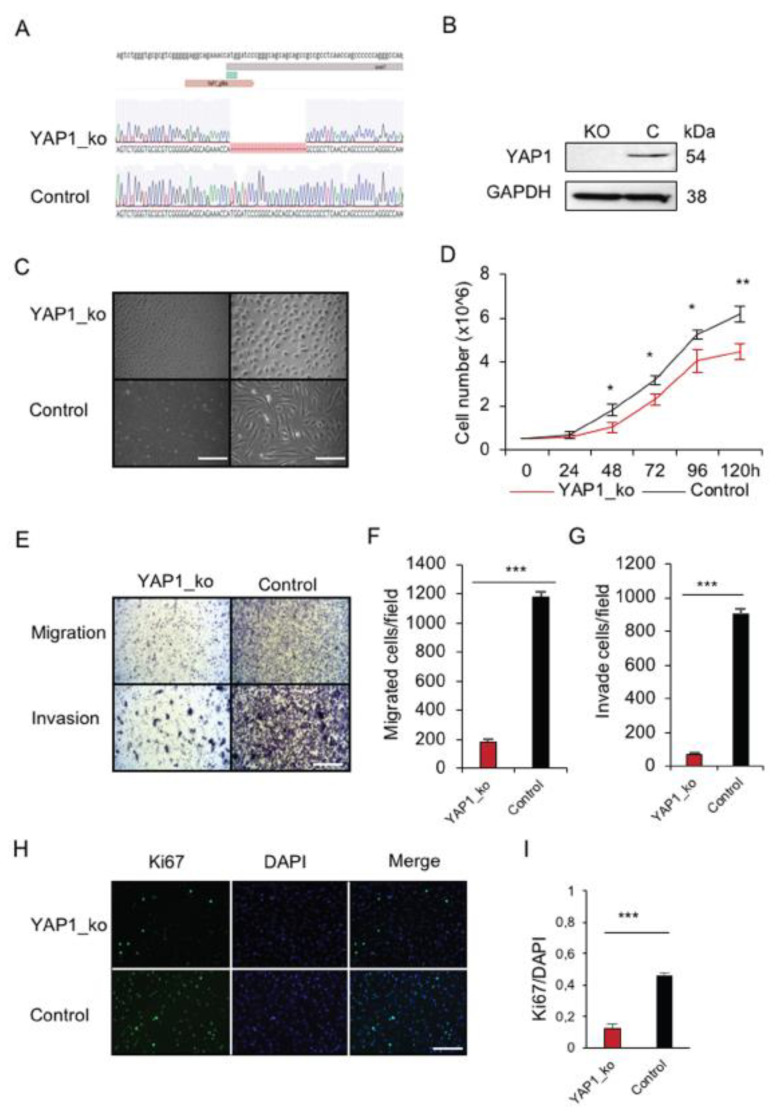
In vitro functional analysis of YAP1 deficiency in p53 deficient primary osteosarcoma cells. (**A**) Sequence analysis showing the result of CRISPR/Cas9 editing of *YAP1* in pig OS cells. (**B**) Western blot showing the lack of YAP1 protein in the edited *flTP53^R167H^* OS cells. (**C**) Representative microscopic view showing the morphology of *YAP1^−/−^/flTP53^R167H^* OS cells. As a control, *flTP53^R167H^* OS cells were transfected with the GFP control vector (left scale bars, 400μm; right scale bars, 200 μm) (**D**) Proliferation result for *YAP1^−/−^/flTP53^R167H^* and *flTP53^R167H^* OS cells. (**E**) Representative microscopic images showing a difference in migration and invasion between *YAP1^−/−^/flTP53^R167H^* and *flTP53^R167H^* OS cells (scale bars, 200 μm). Quantitative measurement of migration (**F**) and invasion (**G**). (**H**) Immunofluorescence staining for Ki67 and DAPI in *YAP1^−/−^/flTP53^R167H^* and *flTP53^R167H^* OS cells. (**I**) Quantification rates of the Ki67 positive cells. * *p* < 0.05, ** *p* < 0.01, *** *p* < 0.001.

**Figure 4 cancers-13-01364-f004:**
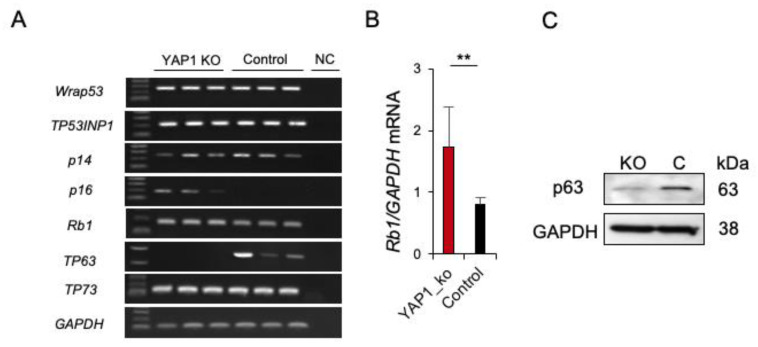
Expression of p53 related genes in *YAP1^−/−^/flTP53^R167H^* OS cells. (**A**) RT-PCR result for *WRAP53,*
*TP53INP1, p14, p16, RB1, TP63, TP73* in *YAP1^−/−^/flTP53^R167H^* and *flTP53^R167H^* OS cells. Three independent transfections for each expression vector were performed. NC—negative control. (**B**) Quantitative RT-PCR of *p16* mRNA expression. *GAPDH* mRNA expression was used as a reference. ** *p* < 0.01. (**C**) Western blot showing lack of p63 expression in *YAP1^−/−^/flTP53^R167H^* OS cells.

**Figure 5 cancers-13-01364-f005:**
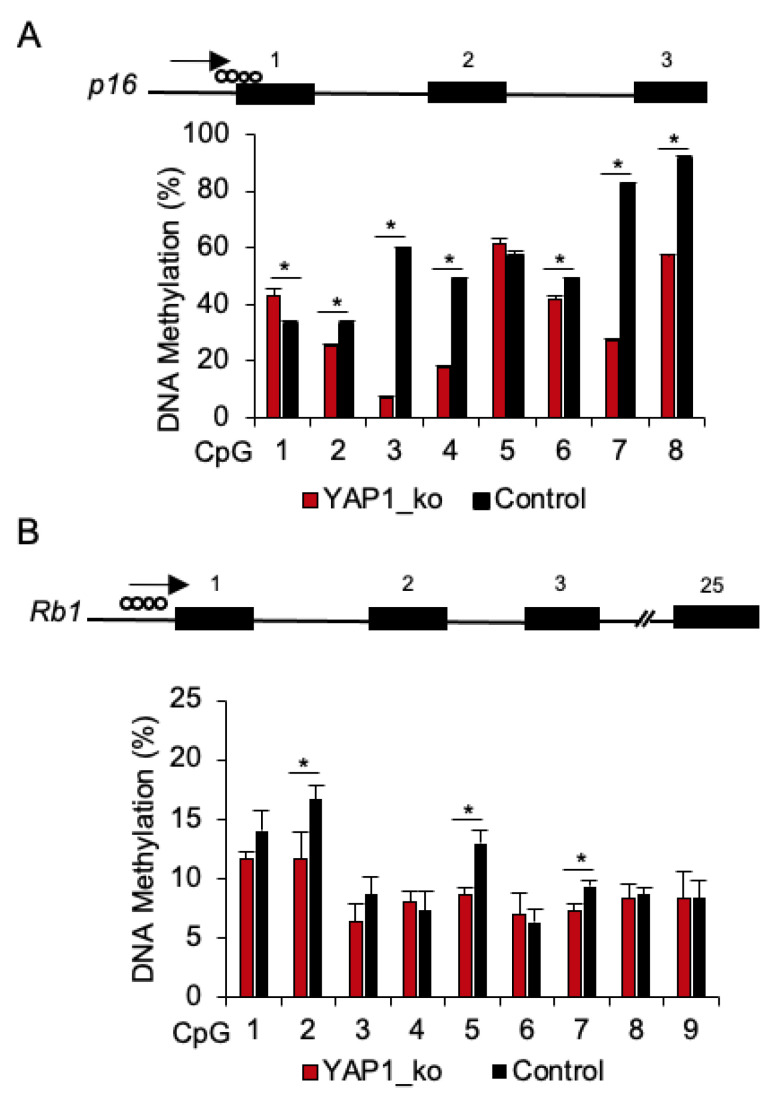
DNA methylation analysis of the p16 and Rb1 promoter regions in *YAP1^−/−^/flTP53^R167H^* OS cells. (**A**) Pyrosequencing result at 8 CpG sites in the *p16* promoter region in *YAP1^−/−^/flTP53^R167H^* (*n =* 3) and *flTP53^R167H^* (*n =* 3) OS cells. (**B**) Pyrosequencing result at 9 CpG sites in the *Rb1* promoter region in *YAP1^−/−^/flTP53^R167H^* (*n =* 3) and *flTP53^R167H^* (*n =* 3) OS cells. * *p* < 0.05.

**Table 1 cancers-13-01364-t001:** The DNA variants showing the most significant allelic expression imbalance in OS.

Chr	Position	Gene	Allele		Healthy Bone		Osteosarcoma		*p* Value
			Ref	Alt	Het No	Proportion	Het No	Proportion	
6	13574601	*SF3B3*	T	C	2	0.55	4	0.32	2.73 × 10^−19^
6	18911317	*HERPUD1*	C	T	2	0.45	4	0.69	1.88 × 10^−41^
6	94787078	*RRAGC*	A	T	3	0.74	5	0.42	5.38 × 10^−21^
9	33085822	*TMEM123*	A	AG	3	0.34	4	0.78	1.11 × 10^−43^
9	33045115	*BIRC3*	T	G	3	0.36	6	0.71	2.17 × 10^−31^
9	33044172	*BIRC3*	G	A	3	0.41	5	0.78	2.14 × 10^−39^
14	72198674	*SRGN*	AT	A	2	0.27	4	0.43	4.52 × 10^−29^
16	71982561	*TNIP1*	T	G	2	0.46	4	0.31	3.74 × 10^−27^

## Data Availability

Data is contained within the article or Appendix A.

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
