# Peer review of "Allelic Expression Imbalance Analysis Identified YAP1 Amplification in p53- Dependent Osteosarcoma"

_cancers, 2021, doi:10.3390/cancers13061364_

Round 1
Reviewer 1 Report
The study I'm here to review describes the involvement of YAP1 in the progression of bone cancer (osteosarcoma, OS) in pigs that develop OS at high frequency, due to a mutated TP53 gene.
The role of YAP in OS progression is already known, however, the novelty of this study has been to have provided the mechanisms by which this happens. Therefore I think this study must be accepted as reflects the high requirements needed for publication on Cancers.
I have just one minor concern:
In paragraph 3.4, the authors write: "Further, the effect of YAP1 deficiency on the expression of tumorigenesis-related genes (Rb1, WRAP53, TP53INP1, p14, p16, TP63, TP73) was studied. RT-PCR and western blot analyses revealed upregulation of p16 and Rb1 (2- fold) and downregulation of TP63 in YAP1-/-/flTP53R167H compared to flTP53R167H pig OS cells (Figure 4A, B).". Actually, in Figures 4A and B no western blots are shown. A western blot related to the expression of only p63 is shown in Figure 4C. Please check and correct the reported sentence, or add new figures, accordingly.
Author Response
We would like to thank the Reviewer for their time and positive comments, which helped us to improve and clarify our manuscript.
In paragraph 3.4, the authors write: "Further, the effect of YAP1 deficiency on the expression of tumorigenesis-related genes (Rb1, WRAP53, TP53INP1, p14, p16, TP63, TP73) was studied. RT-PCR and western blot analyses revealed upregulation of p16 and Rb1 (2- fold) and downregulation of TP63 in YAP1-/-/flTP53R167H compared to flTP53R167H pig OS cells (Figure 4A, B).". Actually, in Figures 4A and B no western blots are shown. A western blot related to the expression of only p63 is shown in Figure 4C. Please check and correct the reported sentence, or add new figures, accordingly.
Response:
We rephrased the sentence as follows:
RT-PCR and qRT-PCR analyses revealed upregulation of p16 and Rb1 (2- fold) and downregulation of TP63 in YAP1-/-/flTP53R167H compared to flTP53R167H pig OS cells (Figure 4A, B). The downregulation of p63 was confirmed by Western blot (Figure 4C).
Reviewer 2 Report
Both human and mouse OS express high level of Yap1 that is amplified in various cancers. YAP1 increases proliferation and chemoresistance of OS. Knockdown of YAP1 inhibits tumor progression. PMID: 24993351; PMID: 27206784.
The current study reports high levels of AEI for SNPs in BIRC3-YAP1 in pigs carrying TP53 mutations. YAP1 amplification is associated with the AEI and progression of OS. Inactivation of YAP1 inhibits proliferation, migration, and invasion, and leads to the silencing of TP63 and reconstruction of p16 expression correlated with lower methylation of its promoter. The authors conclude that the study provides molecular evidence for the role of YAP1 amplification in the progression of p53- dependent OS.
Comments:
As YAP1 amplification is already reported in p53- dependent OS in previous studies, PMID: 24993351, the authors conclusion needs to be reworded to better reflect the significance of the current work.
Other points:
Figure 1A
- Color codes are missing
- the variables (copy number) need to be plotted on the x-axis. Logarithmic scales may be more revealing
Figure 1B
- What is the dimension of the size measurement.
- The differential YAP1 expression was not detected in RNA sequencing, and the authors explained discrepancy by the cutoff of 2- fold threshold. This could be confirmed by examining the RNA sequencing data to see if the values are indeed higher but lower than 2 folds.
Figure 2A
- Some high copy numbers (the dots adjacent to the y-axis) are not associated with increased levels of expression. Please explain or discuss.
- Similarly some high levels of allele expression or copy numbers are not associated with large tumors (B & C). Please discuss.
- E, Only four OS samples are presented. Are there OS that are not expressing high high levels of YAP1?
Figure 4
- A, please define NC. What are the Rb1, P16 expression levels like in non-malignant tissue?
Author Response
We would like to thank the Reviewer for their time and positive comments, which helped us to improve and clarify our manuscript.
Point by point responses:
As YAP1 amplification is already reported in p53- dependent OS in previous studies, PMID: 24993351, the authors conclusion needs to be reworded to better reflect the significance of the current work.
We have rewritten the final sentence in conclusions as follows: "This study confirms the importance of p53/YAP1 network in cancer."
Other points:
Figure 1A
- Color codes are missing
We assume the Reviewer means Figure 2A.
We added the following information:
Gray and red points show expression of A allele in bone and OS samples, respectively. Blue points show the measurements in wild-type samples.
- the variables (copy number) need to be plotted on the x-axis. Logarithmic scales may be more revealing
Figure 1B shows the statistical significance of allelic expression imbalance, not copy number variation, in osteosarcoma samples. We believe that the manhattan plot is presented in a state-of-art format.
Figure 1B
- What is the dimension of the size measurement.
We added info that the tumour size ranged between 1.5 and 18 cm.
- The differential YAP1 expression was not detected in RNA sequencing, and the authors explained discrepancy by the cutoff of 2- fold threshold. This could be confirmed by examining the RNA sequencing data to see if the values are indeed higher but lower than 2 folds.
In the original submission, we mentioned that the fold change of 1.8. However, to make this statement clearer, we have rewritten this as follows:
"A detailed analysis of RNA sequencing data showed only 1.8- fold expression difference of YAP1 AEI between the OS and healthy bone samples, which was below the 2- fold threshold applied."
Figure 2A
- Some high copy numbers (the dots adjacent to the y-axis) are not associated with increased levels of expression. Please explain or discuss.
Yes, this is true mainly for healthy tissues and small OS tumours.
- Similarly some high levels of allele expression or copy numbers are not associated with large tumors (B & C). Please discuss.
Yes, this is true. We think that this is mainly due to the heterocellular nature of the OS tumours and the tissue sections analysed.
- E, Only four OS samples are presented. Are there OS that are not expressing high high levels of YAP1?
In total, we analysed eight samples for YAP1 protein expression by Western blot.
All showed a considerably higher YAP1 expression. Because it was so obvious we decided to show only a representative blot.
Figure 4
- A, please define NC. What are the Rb1, P16 expression levels like in non-malignant tissue?
NC -negative control.
The levels of Rb1 and p16 in non-tumour samples were as shown in the control panel; there is a lack of p16 and a moderate p63 expression. This data is consistent with findings in humans and mice.